# End-to-End Calcification Distribution Pattern Recognition for Mammograms: An Interpretable Approach with GNN

**DOI:** 10.3390/diagnostics12061376

**Published:** 2022-06-02

**Authors:** Melissa Min-Szu Yao, Hao Du, Mikael Hartman, Wing P. Chan, Mengling Feng

**Affiliations:** 1Department of Radiology, Wan Fang Hospital, Taipei Medical University, Taipei 116, Taiwan; manseiyiu@gmail.com (M.M.-S.Y.); ephfm@nus.edu.sg (M.F.); 2Department of Radiology, School of Medicine, College of Medicine, Taipei Medical University, Taipei 110, Taiwan; 3Saw Swee Hock School of Public Health, National University of Singapore, Singapore 117549, Singapore; mikael_hartman@nuhs.edu.sg; 4National University Health System, Singapore 119228, Singapore; 5Department of Surgery, Yong Loo Lin School of Medicine, National University of Singapore, Singapore 117549, Singapore; 6Medical Innovation Development Center, Wan Fang Hospital, Taipei Medical University, Taipei 116, Taiwan; 7Institute of Data Science, National University of Singapore, Singapore 117602, Singapore

**Keywords:** artificial intelligence, calcifications, graph convolution network, deep learning, mammography

## Abstract

**Purpose**: We aimed to develop a novel interpretable artificial intelligence (AI) model algorithm focusing on automatic detection and classification of various patterns of calcification distribution in mammographic images using a unique graph convolution approach. **Materials and methods**: Images from 292 patients, which showed calcifications according to the mammographic reports and diagnosed breast cancers, were collected. The calcification distributions were classified as diffuse, segmental, regional, grouped, or linear. Excluded were mammograms with (1) breast cancer with multiple lexicons such as mass, asymmetry, or architectural distortion without calcifications; (2) hidden calcifications that were difficult to mark; or (3) incomplete medical records. **Results**: A graph-convolutional-network-based model was developed. A total of 581 mammographic images from 292 cases of breast cancer were divided based on the calcification distribution pattern: diffuse (*n* = 67), regional (*n* = 115), group (*n* = 337), linear (*n* = 8), or segmental (*n* = 54). The classification performances were measured using metrics including precision, recall, F1 score, accuracy, and multi-class area under the receiver operating characteristic curve. The proposed model achieved a precision of 0.522 ± 0.028, sensitivity of 0.643 ± 0.017, specificity of 0.847 ± 0.009, F1 score of 0.559 ± 0.018, accuracy of 64.325 ± 1.694%, and area under the curve of 0.745 ± 0.030; thus, the method was found to be superior compared to all baseline models. The predicted linear and diffuse classifications were highly similar to the ground truth, and the predicted grouped and regional classifications were also superior compared to baseline models. The prediction results are interpretable using visualization methods to highlight the important calcification nodes in graphs. **Conclusions**: The proposed deep neural network framework is an AI solution that automatically detects and classifies calcification distribution patterns on mammographic images highly suspected of showing breast cancers. Further study of the AI model in an actual clinical setting and additional data collection will improve its performance.

## 1. Introduction

Breast cancer is the most common cancer among women worldwide; new cases diagnosed in 2018 were estimated at 2.3 million [1]. It is the fifth-most common cause of cancer death in women, accounting for 0.68 million (6.7%) of all new cancer deaths. Early detection and treatment remain the best approaches to minimizing mortality from breast cancers.

Regular screening mammograms beginning at 40 years of age can reduce breast cancer mortality in average-risk women by approximately 20% [2]. Compared with biennial screening, annual screening mammograms showed fewer interval cancers, fewer advanced-stage cancers [3], and lower 10-year breast cancer mortality among women at average risk of developing breast cancer [4]. A standard mammogram includes two projections for each breast: the craniocaudal view and the mediolateral oblique view. Most mammograms are now obtained using fully built high-resolution digital images (over 10 million pixels per image), and they are used to reveal abnormalities, the most common of which are masses, calcifications, architectural distortions of breast tissue, and asymmetries between the breasts. Calcifications are one of the most common early signs of breast cancer, which can be determined by calcifications distribution, number, and certain features. Calcifications in both premenopausal and postmenopausal women indicate a high risk for breast cancers [5].

Calcification distributions refer to how calcifications are distributed across a patient’s breast, which is strongly associated with malignancy [6]. Calcification distributions can be described by descriptors including diffuse, regional, cluster, linear, and segmental, ranked by increasing risk of malignant cancer [7]. According to the Breast Imaging Reporting and Data System (BI-RADS), 5th edition, diffuse distribution is typically benign [8]. Regional distribution is associated with a probability of malignancy at around 26% [9]. Linear and segmental distributions are associated with probabilities of malignancy at around 60% and 62%, respectively [9]. Calcifications with cluster distributions need to be evaluated together with the morphology of the calcifications. Determination of calcification distributions and estimation of their risks are essential for diagnosis. A Korean study reported that 53% cases with cluster distributions and 28% cases with segmental distributions were misinterpreted or underestimated by the initial radiologists and confirmed as malignant at follow-ups [10].

Manual reading of mammograms often leads to false-positive/negative results. Blinded double-reading increases the cancer detection rate, but it also increases the recall rate and the false-positive recall rate [11]. In 2018, AI-antari MA et al. performed a study examining an integrated CAD system used for classifying mass in mammography, with an accuracy over 92% [12]. Alejandro Rodriguez-Ruiz et al. collected 2654 mammograms interpreted by 101 radiologists and created an AI 10-score scale to detect the possibility of breast cancer. This study revealed a 17% reduction in radiologist workload when the AI score was 2 [13]. One solution is to develop artificial intelligence (AI) technologies that can assist radiologists in effectively detecting breast cancers in mammograms. Recent reports describe newly developed AI technologies that can detect breast cancer calcifications. Breast cancer calcifications are defined as tiny high-density spots in breast tissue shown on mammograms, and are also known as microcalcifications, with a size from 0.1 mm to 1 mm, without visible masses [14]. Wang et al. [15] used a discrimination classifier model to improve the diagnostic accuracy of microcalcifications using a semi-automated segmentation method to characterize all microcalcifications in a large dataset. Results show a discriminative accuracy of 87.3% when microcalcifications were characterized alone compared to 85.8% when a support vector machine was employed. Rehman et al. [16] proposed a system called “Intelligent System for Detection of Micro-Calcification in Breast Cancer”, which comprises three primary stages and has an overall classification accuracy of 95.6%. Mayo et al. [17] conducted a retrospective study comparing AI-based detection to computer-aided detection of calcifications in mammographic images, and the false-positive rate was 83% lower in the former. Cai H et al. developed a CNN model for image analysis and classification of mammographic calcifications with a precision of 89.32% and sensitivity of 86.89% [18]. Zobia Suhail et al. developed a novel method using Fisher’s linear discriminant analysis approach combined with an SVM variant, collecting 288 regions of interest (ROIs)x, which resulted in 139 malignant and 149 benign regions, with an accuracy of 96% [19]. However, these technologies were unable to annotate calcifications by distribution pattern, shape, or size using the lexicons/descriptors of the Breast Imaging Reporting and Database System, resulting in a high incidence of false positives.

An automatic computer-aided detection and diagnosis (CADx) system for calcification distributions could greatly reduce radiologists’ time and efforts and help radiologists accurately determine the descriptors. In this study, we adopted an emerging method, graph convolutional networks (GCNs), for distribution classification in mammography images. GCNs apply convolution operations in graphical data to learn visual features and geometrical representations. Many researchers have developed GCN-based applications for mammography images. Liu et al. introduced pseudo-landmarks and graph node mappings to represent regions in mammograms with graphs, and then utilized GCNs to model the multi-view information from these graphs to detect masses [20]. Zhang et al. combined a GCN with convolutional neural networks (CNNs) as a data augmentation model for the detection of lesions in mammograms. The proposed model demonstrated improved performance using the state-of-the-art lesion detection methods, proving itself as an effective data augmentation approach [21].

Though many GCN applications for mammography images have been proposed, and a certain degree of success has been achieved, no existing methods have addressed the classification of calcification distribution patterns. To fill in this gap, we developed a novel graph-convolutional-network-based framework for the automatic detection of calcifications and classification of mammographic images by calcification distribution patterns. This study substantially differs from our previous study [22] in two aspects: firstly, this study introduces an end-to-end approach, rather than focusing on the modeling of annotated calcifications; second, this study introduces a different visualization method to interpret GCN results to help radiologists identify distribution patterns.

## 2. Materials and Methods

### 2.1. Patients and Datasets

The Taipei Medical University Joint Institutional Review Board approved this study (approval number N202006039). Informed consent was waived because of its retrospective nature. All methods were performed in compliance with relevant guidelines and regulations.

In this study, we focused on developing an algorithm to detect and classify calcifications in breasts using a unique deep learning tool. We retrospectively reviewed a set of mammograms diagnosed with documented calcifications in the original radiological reports. All mammograms showed histologically confirmed breast cancers, and all were performed at a single institution between June 2010 and October 2018. All examinations were performed as referrals from breast clinics or healthcare screening centers, and each calcification distribution pattern could be classified as diffuse, segmental, regional, grouped, or linear. Excluded were mammograms with (1) breast cancers with multiple lexicons, such as mass, asymmetry, or architectural distortion without calcifications; (2) hidden calcifications that were difficult to mark; and (3) incomplete medical records. In each of the 292 cases, the two standard views were taken using a stationary unit (Mammomat Inspiration; Siemens, Erlangen, Germany).

### 2.2. Annotation and Ground Truth

The two standard mammographic images of the cancer-positive side were extracted for each patient, and each was annotated with one distribution label. One senior radiological technologist with 15 years of experience in mammography marked calcifications with a large circular region of interest and annotated the morphology as one of five calcification distribution patterns (diffuse, segmental, regional, grouped, or linear) with descriptors of suspicious features (not included in this study). Examples of the annotations are shown in Figure 1. Two additional radiologists (with 12 and 20 years of breast imaging experience) carefully reviewed all labeled images in both subsets and confirmed the annotations in a joint meeting, reaching a consensus on interpretation in all cases. The confirmed annotations were considered the “ground truth” in the dataset.

### 2.3. Study Design

We established an end-to-end framework to classify the distribution of calcifications on mammograms. We developed a method to map mammography images to graph structured data and converted the classification task of classifying distribution of calcifications into a graph classification task. Detailed contributions of this method are listed as below:

We designed a graph construction module which detects calcifications and transforms them into nodes in graphs. We defined node features and an adjacency matrix to represent the calcification graph. Node features were represented by deep feature maps from a convolutional neural network which was trained on patches that were centered at proposed calcifications. Adjacency matrix of nodes was defined according to the spatial relationship among calcifications.We developed a graph convolutional neural network to fuse the node characteristics of the calcification graph and the spatial topological relationship to perform the graph classification task. The graph convolutional neural network was trained to fuse the features and topological structures from neighboring nodes and extract the most correlated information for the classification task.Our developed model is interpretable by highlighting important nodes in graphs. For each distribution descriptor, the highlighted nodes are consistent with the clinical descriptions.

The proposed framework is illustrated in Figure 2.

### 2.4. Calcification Identification

The calcification module was designed to detect calcifications within the regions of interest that were marked on the ground truth images. The first step in this process was to segment the breast region by removing the background and reducing the dimension of the image. In a digital mammogram, the border of the breast area is obvious and distinguishable from the background. Otsu segmentation [23] was employed and resulted in acceptable segmentation. Then, a calcification detection function $f_{detect}$ was introduced to determine the locations of calcifications. A morphological contrast enhancement method [24] was employed. Two morphological operations, the top-hat and bottom-hat transforms, were applied. The former is a residual filter that can retain those features in the image that can be placed in the structural elements while deleting those features that cannot be included. In other words, the top-hat transform is used to segment objects with various brightnesses from the surrounding background in an image with uneven background intensity. It is defined as:(1)$$\begin{array}{c} I_{T}=I-\left[\left(I⊖SE\right)⊕SE\right],\\ f_{detect}\left(x,\;y\right)=\left\{\begin{array}{c} 1,\;\;if\;I_{T}\left(x,\;y\right)\geq T\\ 0,\;\;otherwise \end{array}\right. \end{array}$$
where *I* represents the mammographic image as input, $I_{T}$ is the transformed image, SE is the structuring element, ⊖ and ⊕ represent the morphological erosion operation and morphological dilation operations, respectively, and $\left[\left(I⊖SE\right)⊕SE\right]$ represents the morphological opening operation. $f_{detect}$ returns the thresholding mask of the transformed image, $I_{T}$, regarding the selected threshold, T.

A convolutional neural network (CNN) following $f_{detect}$ was also trained as a means of improving the accuracy of calcification detection. The calcification detection CNN was trained with calcification labels to enhance feature extraction capabilities. The CNN was trained by the loss function defined as:(2)$$L\left(\Theta ,\;\phi \right)=\overset{N}{\underset{i=1}{\sum }}\left(y_{cals}^{\left(i\right)}\log\left(f_{CNN}\left(M_{i}\right)\right)+\left(1-y_{cals}^{\left(i\right)}\right)\log\left(1-f_{CNN}\left(M_{i}\right)\right)\right),$$
where Θ represents the parameters of the convolutional layers in the CNN, ϕ represents the parameters of the fully connected layers, $f_{CNN}\left(\cdot \right)$ represents the forward propagation of the CNN network, $X_{i}$ is a patch image centered at ith detected calcification, and $y_{cals}^{i}$ is the ground truth label of any detected calcification:$$y_{cals}^{\left(i\right)}=\left\{\begin{array}{l} 1\;\mathrm{if}\;M_{i}\;\mathrm{contains}\;\mathrm{true}\;\mathrm{calcifications}\\ 0\;otherwise \end{array}\right.$$

Following Equation (2), we defined the centers of detected calcifications as $S=\left\{\left(x_{1},y_{1}\right),\;\left(x_{2},y_{2}\right),\;\ldots ,\;\left(x_{i},\;y_{i}\right)\right\}$ and a set of patches as $S=\left\{\left(x_{1},y_{1}\right),\;\left(x_{2},y_{2}\right),\;\ldots ,\;\left(x_{i},\;y_{i}\right)\right\}$. For any detected calcification i, $\left(x_{i},y_{i}\right)$ represents the coordinates of its center, and $M_{i}$ represents an image patch of dimension D×D. The coordinate set S and patch set P were then fed into the graph construction module.

### 2.5. Calcification Graph Construction

#### 2.5.1. Learning Feature Representations and Spatial Embeddings of Calcification Patches

To begin construction of the calcification graph, the spatial embeddings and deep feature maps were extracted from calcification patches as node features. Generating visual representations of the calcification patches was a crucial step in this framework. We used CNN to extract feature maps from the intermedia convolutional layers as representations of high-level features. Feature extraction from calcification patches can be represented by:(3)$$X_{patch}^{i}=f_{FE}\left(M_{i};\Theta \right),$$
where $f_{FE}$ is the feature extraction process of a trained CNN with the parameters Θ and $X_{patch}^{i}$ indicating the deep feature maps of patch image $M_{i}$.

Spatial information is crucial for identifying calcification distribution. To preserve the spatial information, a linear transformation was used to transform the spatial information of the calcification into a learnable embedding:(4)Xemb=S· Wemb+bemb,
where S=x1,y1,x2,y2,…,xi,yi indicates the set of center coordinates of the proposed calcifications; Wemb∈ℝDemb×N represents the learnable weights; and bemb represents biases.

#### 2.5.2. Building a Calcification Graph

A calcification graph was built for each mammographic image to map the input image to graph-structure data. Graph-structure data were defined by the node feature matrix X∈ℝN×D and the adjacency matrix X∈ℝN×D, where N represents the number of nodes, and D represents the dimension of the node features. In this framework, X was constructed based on the fusion of features that were extracted from the CNN and spatial embeddings. The learned features from the calcification patches and spatial embeddings were fused into node features in the calcification graph:(5)Xnode=Xpatch ‖ Xemb,
where ‖ indicates the concatenation operation.

The adjacency matrix A was constructed from the spatial relationships between proposed calcification regions. The matrix was obtained by constructing a k-nearest-neighbor graph [25] using the Cartesian distance between nodes. In other words, given a distance function $d\left(\cdot ,\cdot \right)$, if $d\left(x_{i},x_{j}\right)$ was among the k neighbors at the smallest distance from $x_{i}$ to all other nodes, then $A_{i,j}=1$, otherwise $A_{i,j}=0$.

#### 2.5.3. Learning Spatial and Distribution Relationships Using a Graph Neural Network

A dynamic-edge-conditioned convolution network [26] was used to design the graph convolution layers so that node feature embeddings in the graph structure could be learned. The dynamic-edge-conditioned convolution is performed in a spatial domain where filter weights are conditioned on edge attributes and are dynamically generated for each specific input. Applications in graph classifications in point clouds have demonstrated the usefulness of encoding edge attributes. The convolution operation at layer l can be formulated as:(6)$$X^{l}\left(i\right)=\frac{1}{\left|N\left(i\right)\right|}\overset{\;}{\underset{j\;\in N\left(i\right)}{\sum }}F^{l}\left(L\left(j,i\right);\;w^{l}\right)X^{l-1}\left(j\right)+b^{l},$$
where $F^{l}\left(\cdot \right)$ represents a feed-forward network that encodes edge labels $L\left(j,\;i\right)$ into edge-specific embeddings, $N\left(i\right)$ represents the set of neighboring vertices of node i, and $b^{l}$ represents a learnable bias. Though multiple graph convolutional layers, node feature $X^{l}\left(i\right)$ fuses the visual and topological information from multi-hop neighboring nodes and extracts the most information that is relevant to the distribution class. Following the graph convolutional layers, the node feature matrix was flattened to obtain a finite-dimensional vector as a global representation of the calcification graph:(7)$$x=\mathrm{flatten}(X_{GAT}),$$
where $X_{GAT}$ is the input node feature matrix, and x is the output feature vector.

In the last step, a fully connected layer with a sigmoid activation function was added as a classifier to perform the graph classification task. The output of the last fully connected layer was computed to represent the classification probability of calcification distribution:(8)$$\hat{y}=\sigma \left(xW_{fc}+b_{fc}\right),$$
where σ represents the sigmoid activation function, and $W_{fc}$ and $b_{fc}$ represent the learnable matrix and bias, respectively, of the fully connect layer.

Classification of the calcification distribution patterns suffers from class imbalance, but this can be addressed utilizing focal loss [27] to train the proposed model:(9)$$L\left(p_{t}\right)=-{\left(1-p_{t}\right)}^{\gamma }\log\left(p_{t}\right),$$
where γ represents the focusing parameter. When γ=0, $L\left(p_{t}\right)$ is equivalent to cross entropy loss. $p_{t}$ represents the model-estimated probability. $p=p_{t}$ if the one-hot class label y=1, otherwise $p_{t}=1-p$. If the example is misclassified, ${\left(1-p_{t}\right)}^{\gamma }$ tends to 1; if the example is correctly classified, ${\left(1-p_{t}\right)}^{\gamma }$ tends to 0. Therefore, the loss contribution of easy examples from majority classes are down-weighted while the loss contribution of difficult examples from minority classes remain the same as cross entropy loss. During training, focal loss thus reduces the bias towards classifying the cases into majority distribution classes.

### 2.6. Data Analysis

For comparison, several existing methods were used to classify the images into calcification distribution patterns: ResNet [28], DenseNet [29], MobileNet [30], and EfficientNet [31]. These methods demonstrated the state-of-the-art performances on image classifications and have been widely adopted in various applications tasks:

ResNet was proposed by He et al. and won the ILSVRS competition in 2015. The authors proposed residual blocks with skip connections to train deep CNNs with up to 152 layers. ResNet is one of the most popular and successful methods in the computer vision community with various applications in medical imaging [32,33,34]. ResNet-50 was used in this study.DenseNet was proposed with dense connections between layers to reuse features and preserve the global state. DenseNet demonstrated outperforming results on small benchmarking datasets such as Cifar-10 and Cifar-100 [29]. DenseNet-121 was adopted in this study.MobileNet was proposed primarily with depthwise separable convolutions as an efficient model with high accuracy and low latency for mobile and embedded applications. The effectiveness of MobileNets has been demonstrated for various applications, such as object detection, traffic density estimation, and computer-aided diagnosis systems [30,35,36,37,38]. MobileNetV2 was used in this study.EfficientNets is a family of models proposed by Google in 2019. EfficientNets outperformed state-of-the-art accuracy with up to 10 times better efficiency. A compound scaling method was proposed in EfficientNets to expand the depth, width, and resolution of the network. EfficientNets obtained state-of-the-art capacity in various benchmark datasets while requiring less computing resource than other models. EfficientNet-B7 was used in experiments of this study.

All baseline models were trained using balanced sampling from each class over 100 epochs. Model training and testing were implemented using PyTorch and OpenCV on an Ubuntu server with four Tesla V100 graphics processing units. The mean (±standard deviation) performance of each method was evaluated using Scikit Learn with Python 3.6. Classification performances were evaluated based on precision, sensitivity (recall), specificity, F1 score, accuracy, and the multi-class area under receiver operating characteristic.

## 3. Results

### 3.1. Implementation Details

MobileNet [30] was adopted for these experiments as the calcification detection CNN; it was trained over 100 epochs. The size of the calcification patches was set to 14×14 pixels. A six-layer CNN was constructed as the feature extractor following ResNet architecture [39]. The extracted features were fused with position embeddings as the node features of the resultant calcification graphs. Hyperparameters were selected by grid search. The dimension of node features is 80. The number of graph convolutional layers was set to three, and the number of fully connected layers was set to two. The number of neighbors in KNN was selected to be eight. γ was set to two, the same as the original focal loss paper. The network was trained using the Adam [40] optimizer over 100 epochs, and the learning rate was initially set to $3\times 10^{-4}$ and decayed during the training process.

### 3.2. Results

We trained and tested the proposed GNN and baseline models in a 5-fold cross validation manner. In each split, 80% of the 292 breast cancer cases were used for training, and 20% cases were used to evaluate the results. Calcifications following the cluster or regional distribution patterns were the most common, accounting for 50.1% and 27.7% of the samples, respectively. The segmental pattern was found in 14.2%, the diffuse pattern in 6.0%, and the linear pattern in 2.0%. In this study, the total number of patients was *n* = 199, and the mean age was 54 ± 11.90. The reasons for mammography examinations were diagnosis (*n* = 147, 73.5%) and screening, (*n* = 53, 26.5%). The majority revealed palpable mass (*n* = 99, 49.5%), followed by asymptomatic findings or subsequent screening (*n* = 51, 25.5%); lumps (*n* = 25, 12.5%); positive mammographic findings referred from outside hospitals (*n* = 16, 8%); and miscellaneous (*n* = 9, 4.5%). According to the standard of the BI-RADS breast density and final assessment, breast densities were classified into composition A (*n* = 5, 2.5%); B (*n* = 34, 17%); C (*n* = 144, 72%), and D (*n* = 17, 8.5%). The final category assessment showed 4A, eight cases (4%); 4B, thirty-five cases (17.5%); 4C, thirty-nine cases (19.5%); and 5, one hundred and eighteen cases (59%). After surgical resection and histopathology tissue evaluation, there were one hundred and nineteen cases (59.5%) of invasive ductal carcinoma (IDC); sixty-three cases (31.5%) of ductal carcinoma in situ (DCIS); seven cases (3.5%) of IDC plus DCIS; and eleven cases (5.5%) resulting from another pathology.

Model performance after training was evaluated using the testing subset in each split. The proposed method outperformed the other methods in distribution pattern classification across most performance metrics, including: sensitivity, specificity, F1 score, accuracy, and multi-class area under the curve (AUC). Table 1 shows these metrics for all tested methods, including the proposed method. The confusion matrix of one fold in five-fold cross validation is shown in Figure 3, and receiver characteristic curves (ROCs) of the distribution pattern classification are shown in Figure 4.

We interpreted the prediction results using graphical saliency maps generated by the Grad-Cam [41]. In these saliency maps, node importance is encoded with the color of the node. Figure 5 shows selected visualization examples. For each distribution category, an example mammogram is shown on the left, a graphical saliency map is shown in the middle, and a saliency map generated from the ResNet baseline model is shown on the right for comparison.

## 4. Discussion

To the best of our knowledge, this is the first study on the use of AI technology to automatically classify mammographic images according to calcification distribution patterns. Therefore, we adopted several methods of image classification (ResNet, DenseNet, MobileNet, and EfficientNet) as baseline methods for comparative purposes. This study showed that our proposed method outperformed the others in classifying calcification distribution patterns across all evaluation metrics. For instance, the proposed model yielded a better AUC than the baseline models by a range from 9.9% to 29.1%.

Classifying calcification distribution patterns remains challenging because it is difficult to learn the discriminative semantic representations. By employing a graph convolutional neural network to fuse the node characteristics of the calcification graph and the spatial topological relationships, the graph classification task can be accurately performed. This calcification module can detect calcifications in mammograms, assign them as nodes in a calcification graph, and then represent that graph using the defined node features and adjacency matrix. Node features were represented by deep feature maps from a CNN that was trained on patches centered at proposed calcification locations. The adjacency matrix was defined according to the spatial relationships between calcifications. The proposed model works similar to a real-world radiology practice in which the topological relationship between detection and classification of the calcification distribution pattern is considered in mammographic images.

Conventional CNNs are widely adopted in image analysis because they can exploit shift invariance, local connectivity, and image data compositionality as regular grids in the Euclidean space [42,43]. However, calcifications in mammograms are often not grid-like, and non-local information is needed. This leads to the failure of traditional CNNs, such as those used as baseline models (see Table 1). For example, in a diffuse distribution pattern, calcifications are scattered randomly throughout the breast, but in a regional distribution, they are scattered across a large volume of breast tissue. Distinguishing between the two requires both long-range and non-local information from the mammographic image. By incorporating spatial information, the proposed method can outperform every baseline model in distinguishing between these two distribution patterns. Similar benefits of graph convolutional networks have also been found in semantic segmentation and point cloud classification problems [44,45,46].

Our developed model is interpretable, and the interpretation of predictions is consistent with the clinical knowledge. In Figure 5a, all nodes are of high importance, and they are in a small cluster, leading to identification of the cluster distribution pattern. The important calcifications in Figure 5b were found to form a linear distribution pattern. Figure 5c shows a segmental distribution pattern, where important nodes comprise a large part of the calcifications, and the region is larger than what is seen in the regional distribution pattern (Figure 5d). These coincide with clinical classification criteria. Finally, Figure 5e shows that in the diffuse distribution pattern, important nodes comprise the largest part of the graph, an attribute that agrees with the radiologist’s annotation. In all examples in Figure 5, graph saliency maps from the proposed model are more consistent with the radiologist’s annotations than the saliency maps generated from the baseline ResNet model. In Figure 5a,b,d, the saliency maps generated from ResNet are different from the radiologist’s annotations on the left, which means ResNet does not capture the important visual features for the distribution classification task. In Figure 5c,e, the baseline ResNet model fails to classify the images into the correct distribution classes, such that the activations are low with respect to the correct classes, and the saliency maps of these cases show non-activations in blue. The visualized heat maps show that the proposed model can identify important calcifications and use those to distinguish between various distribution patterns.

This study has several limitation(s). The number of cases was relatively small, given the minor representation of the linear calcification distribution pattern. To resolve this class imbalance, we introduced focal loss during training. Further studies employing larger representations of the linear calcification distribution pattern are needed. Because this is a preliminary study focusing on the novelty of our proposed method and comparing its performance with other options, this preliminary dataset did not include mammograms with benign calcifications or normal mammograms without calcifications. This design feature should be eliminated in further studies. Our ongoing study employs a model designed to classify additional features and distributions of calcifications that can be found on mammograms.

## 5. Conclusions

In summary, we established an end-to-end framework for classifying calcification distribution patterns seen on mammographic images. We developed a method to map mammographic images, graph structured data, and convert the classification task to a graph classification task. Our proposed deep neural network framework is an AI solution that can automatically detect and classify calcification distribution patterns on mammograms that are highly suspected of showing breast cancer. Further studies of the AI model are needed, utilizing more variety in the input data, to improve its performance.

## Figures and Tables

**Figure 1 diagnostics-12-01376-f001:**
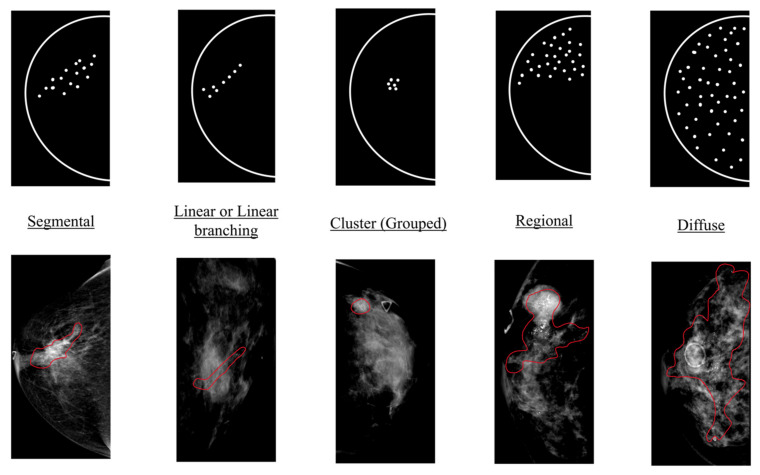
Examples of calcification distribution descriptors. For each distribution descriptor, one illustration is shown on the top and one example from the study dataset is shown at the bottom. Illustrations are adapted according to BI-RADS 5th Edition [9]. Calcification patterns are annotated and marked by red contours in examples.

**Figure 2 diagnostics-12-01376-f002:**
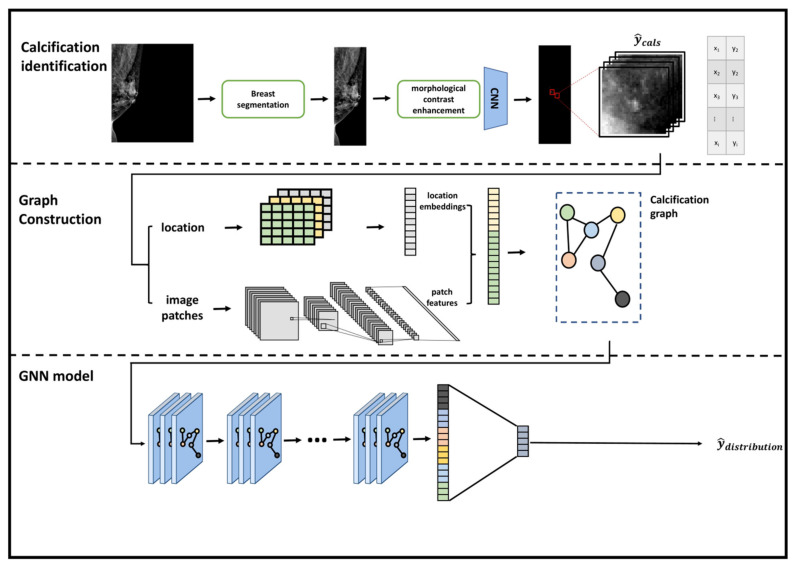
Overview of proposed graph neural network model framework.

**Figure 3 diagnostics-12-01376-f003:**
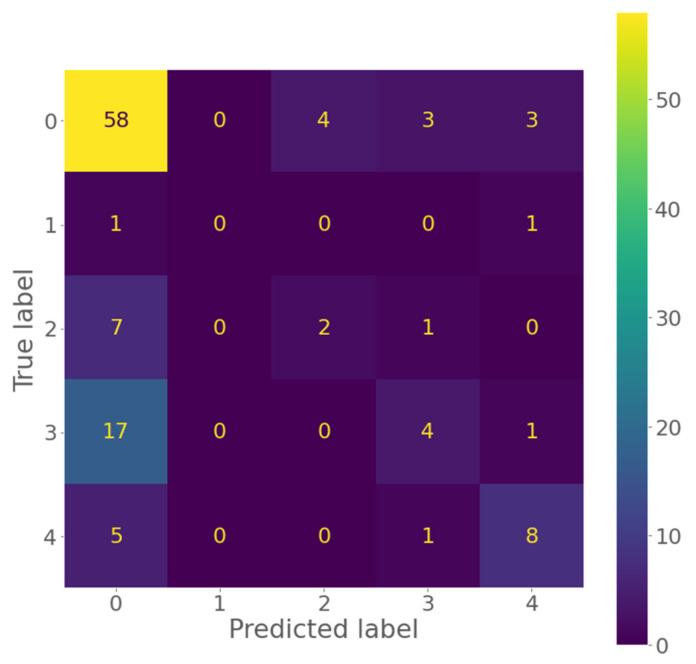
Confusion matrix for distribution classification in one fold of five-fold cross validation.

**Figure 4 diagnostics-12-01376-f004:**
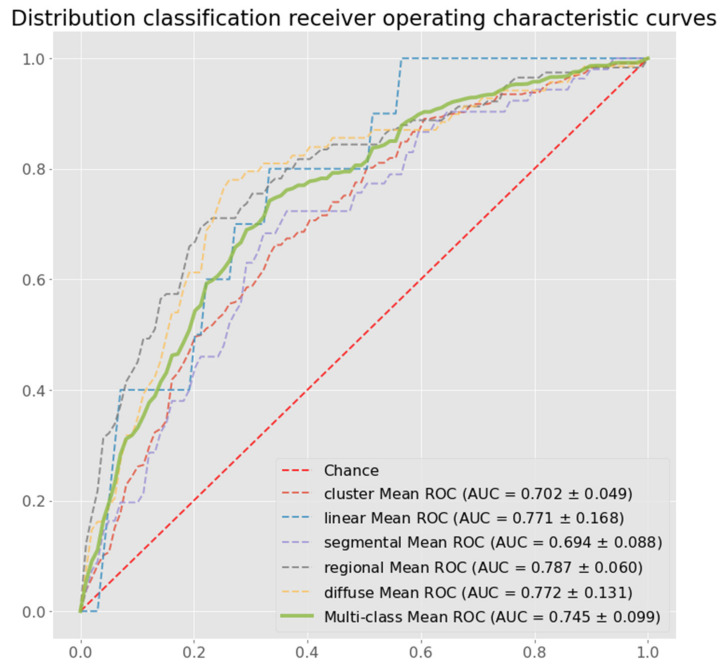
Multi-class receiver operating characteristic curves for distribution classification.

**Figure 5 diagnostics-12-01376-f005:**
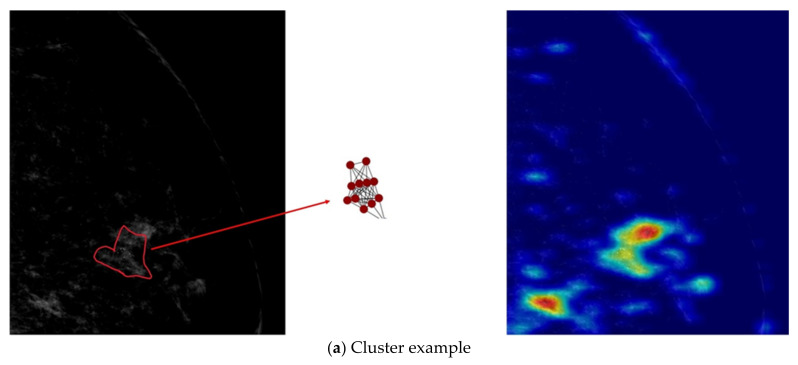
Visualizations of calcification diffusion patterns. One example from each distribution category is selected to be shown in (**a**–**e**). In each left panel, the radiologist’s annotation is outlined in red. Each middle panel shows the graphical saliency map for the corresponding image. Nodes are color-coded according to importance, where blue indicates low importance to the proposed network, yellow indicates medium importance, and red indicates high importance. Each right panel shows the saliency map using the ResNet baseline.

**Table 1 diagnostics-12-01376-t001:** Performance of various methods on overall distribution classification.

	Precision	Sensitivity	Specificity	F1 Score	Accuracy	Multi-Class AUC
ResNet	0.388 (±0.067)	0.594 (±0.019)	0.810 (±0.013)	0.459 (±0.044)	0.594 (±0.019)	0.672 (±0.035)
DenseNet	0.388 (±0.060)	0.590 (±0.013)	0.808 (±0.012)	0.451 (±0.034)	0.590 (±0.013)	0.657 (±0.025)
MobileNet	0.507 (±0.037)	0.607 (±0.009)	0.816 (±0.008)	0.481 (±0.018)	0.607 (±0.009)	0.695 (±0.882)
EfficientNet	0.356 (±0.043)	0.581 (±0.009)	0.802 (±0.003)	0.430 (±0.015)	0.581 (±0.009)	0.695 (±0.030)
**Proposed Method**	**0.522** (±**0.028**)	**0.643** (±**0.017**)	**0.847** (±**0.009**)	**0.559** (±**0.018**)	**0.643** (±**0.017**)	**0.745** (±**0.030**)

Abbreviation: AUC, area under the curve. Bold here is used to highlight the performance of the proposed model.

## Data Availability

The data that support the findings of this study are available from the corresponding author upon reasonable request.

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
