# Peer review of "End-to-End Calcification Distribution Pattern Recognition for Mammograms: An Interpretable Approach with GNN"

_diagnostics, 2022, doi:10.3390/diagnostics12061376_

Round 1

Reviewer 1 Report

Overall:

In this paper is proposed an interpretable method, graph-based, for the classification of micro-calcification distribution. Although the results reported are not good, it is a very interesting method to solve an important problem.

The paper lacks explanation on several points and there is a need for better organization. I describe in detail my concerns below.

The main issues with this work are:

  • The classes are heavily imbalanced, how do the authors address that? The "focal loss" is mentioned in Sec 4, but it is not explained at all.
  • In Sec. #1 is mentioned that calcifications are common for older women. Please put a reference that supports this.
  • The paragraph "Manual reading..." in Sec. #1 seems to be the starting of what I recommend to be a "Related Works" section. Besides that, I recommend including more related works, the review of literature is very small.
  • In Sec. 2.1 the dataset is briefly described. However, in the first paragraph of Sec. 3.2 explains more about the data. I suggest putting it all together.
  • In Sec. 2.3, is mentioned the fusion of node characteristics. I recommend to the authors explain more about how this fusion is done.
  • Sec. 2.4 mentioned a function f_detect, but it never appears in the mentioned equations. I suggest removing it or maybe one equation is missing?
  • Sec. 3.1 "Implementation Details" mentioned only the MobileNet. Maybe this is a text fragment lost in the paper?
  • In Table 1. I suggest to the authors report the accuracy in the range 0-1 , just to keep the uniformity compared to the other measurements.
  • Fig. 4 shows a confusion matrix that is not specified what are the classes being classified. And this figure comes before Fig. 3.
  • Fig. 4.c right side shows an image different from the left one. Moreover, 4.a and 4.e doesn't show any salience map. Is this an error or expected behaviour for these cases? Please, explain better these results.

Author Response

Authors’ Response to Reviews comments

End-to-end calcification distribution pattern recognition for mammograms: an interpretable approach with GNN

Yao et al.

Diagnostics, diagnostics-1727169

  1. Reviewer #1

In this paper is proposed an interpretable method, graph-based, for the classification of micro-calcification distribution. Although the results reported are not good, it is a very interesting method to solve an important problem.

The paper lacks explanation on several points and there is a need for better organization. I describe in detail my concerns below.

RC: The classes are heavily imbalanced, how do the authors address that? The "focal loss" is mentioned in Sec 4, but it is not explained at all.

AR: Thank you for the question. The class imbalance is addressed by focal loss, which is designed to address the scenario in which there is an extreme imbalance between the classes during training. Please refer to Section 2.5.3 and quoted below.

    Classification of the calcification distribution patterns suffers from class imbalance, but this can be addressed utilizing focal loss [21] to train the proposed model:

where represents the focusing parameter. When is equivalent to cross-entropy loss.  represents the model estimated probability.  if the one-hot class label, otherwise, If the example is misclassified,  tends to 1; if the example is correctly classified, tends to 0. Therefore, the loss contribution of easy examples from majority classes is down-weighted while the loss contribution of difficult examples from minority classes remains the same as cross-entropy loss. During training, focal loss thus reduces the bias towards classifying the cases into majority distribution classes.

RC: In Sec. #1 is mentioned that calcifications are common for older women. Please put a reference that supports this.

AR: Thank you for the concern. Mammographic calcifications are the early sign of breast cancer and could be well explained that they have a similar high risk for breast cancer in both premenopausal and postmenopausal women. (the word “older” has been deleted) 

Introduction

Line 100… Calcifications are the common and reliable early signs of breast cancer, which are determined by this calcification's distribution, number, and certain features. Calcifications have a similar high risk for breast cancer in both premenopausal and postmenopausal women. [5]

RC: The paragraph "Manual reading..." in Sec. #1 seems to be the starting of what I recommend to be a "Related Works" section. Besides that, I recommend including more related works, the review of literature is very small.

AR: Thanks for your suggestion. More literatures have been added to the manuscript.

(Manual reading of mammograms…. In 2018, AI-antari MA et al. performed a study of an integrated CAD system used for the classified mass and cancer detection in mammography, with accuracy of more than 92% [12].  Alejandro Rodriguez-Ruiz et al. collected 2654 mammograms and interpreted them by 101 radiologists, creating an AI 10-score scale to detect the possibility of breast cancer. This study revealed a 17%  reduction in radiologist workload when AI scores at 2 [13].

…………………………………………………….. Breast cancer calcifications defined as tiny high density spots in breast tissue shown on mammograms, and are also known as microcalcifications, with sizes measured 0.1mm to 1mm, without visible masses[14]. …………………………………………

………………………………………………………… Cai H et al. developed a CNN model for the images analysis and mammographic calcifications classification with precision of 89.32% and a sensitivity of 86.89% [18]. Zobia Suhail et al. developed a novel method using Fisher Linear discriminant analysis approach in combination with the SVM variant, collecting 288 regions of interest (ROIs) resulting in 139 malignant and 149 benign; with an accuracy of 96% [19].

RC: In Sec. 2.3, is mentioned the fusion of node characteristics. I recommend to the authors explain more about how this fusion is done.

AR: Thank you for your suggestion. We have added explanations of the fusion of node features in Section 2.3 and Section 2.5.3. Detailed explanations are quoted below:

2.3. Study design

  1. We developed a graph convolutional neural network to fuse the node characteristics of the calcification graph and the spatial topological relationship to perform the graph classification task. The graph convolutional neural network is trained to fuse the features and topological structures from neighboring nodes and extract the most correlated information for the classification task.

2.5.3. Learning spatial and distribution relationships using a graph neural network

where represents a feed-forward network that encodes edge labels into edge-specific embeddings,  represents the set of neighboring vertices of node, and represents a learnable bias. Through multiple graph convolutional layers, node feature fuse the visual and topological information from multi-hop neighboring nodes and extract the most information that is relevant information to the distribution class.

RC: Sec. 2.4 mentioned a function f_detect, but it never appears in the mentioned equations. I suggest removing it or maybe one equation is missing?

AR: Thank for you your suggestions.  represents the thresholding function following. We have added the equation into equation (1) and quoted it below:

2.4. Calcification identification

where represents the mammographic image as input,  is the transformed image,  is the structuring element,  and represents the morphological erosion operation and morphological dilation operations, respectively, and represents the morphological opening operation.  returns the thresholding mask of the transformed image, regarding the selected threshold.

RC: Sec. 3.1 "Implementation Details" mentioned only the MobileNet. Maybe this is a text fragment lost in the paper?

AR: Thanks for pointing this out. The implementation detail part was missing in file upload. We have quoted the implementation details below:

MobileNet [24] was adopted for these experiments calcification detection CNN; it was trained over 100 epochs. The size of the calcification patches was set to pixels. A 6-layer CNN was constructed as the feature extractor following ResNet architecture [33]. The extracted features were fused with position embeddings as the node features of the resultant calcification graphs. The dimension of node features is 80. The number of graph convolutional layers was set to 3, and the number of fully connected layers was set to 2. The network was trained using the Adam [34] optimizer over 100 epochs, and the learning rate was initially set to and decayed during the training process.

RC: In Table 1. I suggest to the authors report the accuracy in the range 0-1, just to keep the uniformity compared to the other measurements.

AR: Thank you for your suggestion. We have re-scaled the reported accuracy to the range 0-1 in Table 1. The revised table is quoted as below:

Table 1. Performance of various methods on overall distribution classification.

Precision

Sensitivity

Specificity

F1 Score

Accuracy

Multi-class AUC

ResNet

0.388 (±0.067)

0.594 (±0.019)

0.810 (±0.013)

0.459 (±0.044)

0.594 (±0.019)

0.672 (±0.035)

DenseNet

0.388 (±0.060)

0.590 (±0.013)

0.808 (±0.012)

0.451 (±0.034)

0.590 (±0.013)

0.657 (±0.025)

MobileNet

0.507 (±0.037)

0.607 (±0.009)

0.816 (±0.008)

0.481 (±0.018)

0.607 (±0.009)

0.695 (±0.882)

EfficientNet

0.356 (±0.043)

0.581 (±0.009)

0.802 (±0.003)

0.430 (±0.015)

0.581 (±0.009)

0.695 (±0.030)

Proposed Method

0.522 (±0.028)

0.643 (±0.017)

0.847 (±0.009)

0.559 (±0.018)

0.643 (±0.017)

0.745 (±0.030)

RC: Fig. 4 shows a confusion matrix that is not specified what are the classes being classified. And this figure comes before Fig. 3.

AR: Thank you for pointing this out. We have added the displayed class names and moved this figure before figure 3, as shown in below:

Figure 4. Confusion matrix for distribution classification in one fold of 5-fold cross-validation.

RC: Fig. 4.c right side shows an image different from the left one. Moreover, 4.a and 4.e doesn't show any salience map. Is this an error or expected behavior for these cases? Please, explain better these results.

AR: Thanks for your question. We assume your question refers to the negative saliency maps (pure blue-color heatmaps) in 4.b and 4.e. The negative heatmaps from these cases are expected. The reason is that the saliency maps are generated with respect to the true class labels. For cases 4. b and 4. e, the baseline ResNet model fails to classify the images into the correct distribution classes, such that the activations are low with respect to the correct classes. Therefore, the saliency maps of these cases show non-activations in blue colors.

Our developed model is interpretable and the interpretation of predictions is consistent with the clinical knowledge. In Figure 5(a), all nodes are of high importance, and they are in a small cluster, leading to the identification of the cluster distribution pattern. In Figure 5(b), the important calcifications were found to form a linear distribution pattern. Figure 5(c) shows a segmental distribution pattern, where important nodes comprise a large part of the calcifications, and the region is larger than what is seen in the regional distribution pattern (Figure 5(d)). These coincide with clinical classification criteria. Finally, Figure 5(e) shows that in the diffuse distribution pattern, important nodes comprise the largest part of the graph, an attribute that agrees with the radiologist’s annotation. In all examples in Figure 5, graph saliency maps from the proposed model are more consistent with the radiologist’s annotations than the saliency maps generated from the baseline ResNet model. In Figures 5 (a), (b), and (d), the saliency maps generated from ResNet are different from the radiologist’s annotations on the left, which means ResNet does not capture the important visual features for the distribution classification task. In Figures 5 (c) and (e), the baseline ResNet model fails to classify the images into the correct distribution classes, such that the activations are low with respect to the correct classes and the saliency maps of these cases show non-activations in blue colors. The visualized heat maps show that the proposed model can identify important calcifications and use those to distinguish between various distribution patterns.

Reviewer 2 Report

The manuscript proposed an end-to-end framework for calcification distribution pattern classification in mammographic images. A calcification graph was constructed based on the detected calcifications. A graph convolutional neural network which fuses the node characteristics of the calcification graph and the spatial topological relationship was employed to perform graph classification. Evaluation and comparison with other state-of-the-art methods showed the superiority of the proposed framework.

Major comments:

  1. The image processing steps mentioned in Line 169-181 were not plotted in Figure 2. Also, convolutional neural network (CNN) following fdetect was not plotted.
  2. Line 231-235 is a little bit confusing. Is k (neighbors) a fixed parameter for all the nodes? If it is, how to choose it?
  3. What does the colored bar on the top of the right figure of Figure 3 (a)(b)(c) refer to? Is it sort of boundary issue?

Author Response

Authors’ Response to Reviews comments

End-to-end calcification distribution pattern recognition for mammograms: an interpretable approach with GNN

Yao et al.

Diagnostics, diagnostics-1727169

Reviewer #2

The manuscript proposed an end-to-end framework for calcification distribution pattern classification in mammographic images. A calcification graph was constructed based on the detected calcifications. A graph convolutional neural network which fuses the node characteristics of the calcification graph and the spatial topological relationship was employed to perform graph classification. Evaluation and comparison with other state-of-the-art methods showed the superiority of the proposed framework.

RC: The image processing steps mentioned in Line 169-181 were not plotted in Figure 2. Also, the convolutional neural network (CNN) following was not plotted.

AR: Thank you for the comments. We have added the  (morphological contrast enhancement) and CNN in Figure 2.

Figure 2. Overview of proposed graph neural network model framework.

RC: Line 231-235 is a little bit confusing. Is k (neighbors) a fixed parameter for all the nodes? If it is, how to choose it?

AR: Thank you for your question. (number of neighbors in KNN) is a fixed hyperparameter that was selected by grid search. We have added the explanations in Section 3.1 Implementation details, quoted below:

MobileNet [24] was adopted for these experiments calcification detection CNN; it was trained over 100 epochs. The size of the calcification patches was set to pixels. A 6-layer CNN was constructed as the feature extractor following ResNet architecture [33]. The extracted features were fused with position embeddings as the node features of the resultant calcification graphs. Hyperparameters were selected by grid search. The dimension of node features is 80. The number of graph convolutional layers was set to 3, and the number of fully connected layers was set to 2. The number of neighbors in KNN was selected to be 8. and was set to 2, the same as the original focal loss paper. The network was trained using the Adam [34] optimizer over 100 epochs, and the learning rate was initially set to and decayed during the training process.

RC: What does the colored bar on the top of the right figure of Figure 3 (a)(b)(c) refer to? Is it a sort of boundary issue?

AR: Thanks for pointing this out. Yes, the color bar on the top of the right figure of Figure (a), (b), and (c) are caused by image boundary issues. We have cropped the boundary to avoid confusion. 

Round 2

Reviewer 1 Report

Dear authors, thank you for reviewing and answering my questions.

However, I have tho other points:

-  Sec. 1: For the sentence "Calcifications are the common and reliable early signs of 55 breast cancer,". I recommend: "Calcifications are one of the most common early signs of 55 breast cancer,". For example, this is because there is architectural distortion, which is also a sign that could appear even before the calcifications or masses.

- Figure 5.c the original image (left) is still different from the one on the right.

Author Response

Reviewer #1 comments

Sec. 1: For the sentence "Calcifications are the common and reliable early signs of breast cancer," I recommend: "Calcifications are one of the most common early signs of 55 breast cancer,". For example, this is because there is architectural distortion, which is also a sign that could appear even before the calcifications or masses.

RESP: Thank you! The sentence has been revised with track change accordingly. (Introduction, 2nd paragraph)

- Figure 5. c the original image (left) is still different from the one on the right.

RESP: Thank you! The wrong figure has been corrected accordingly.